

# An optimized multi-attribute decision-making approach to construction supply chain management by using complex picture fuzzy soft set

Ali Asghar[1], Khuram A. Khan[1], Marwan A. Albahar[2] and Abdullah Alammari[3]

[1] Department of Mathematics, University of Sargodha, Sargodha, Sargodha, Pakistan
[2] Computer Science Department, Umm Al-Qura University, Mecca, Saudia Arabia
[3] Faculty of Education, Curriculums and Teaching Department, Umm Al-Qura University, Makkah, Saudi Arabia

Corresponding author
Khuram A. Khan,
khuram.ali@uos.edu.pk

## ABSTRACT

Supplier selection is a critical decision-making process for any organization, as it directly impacts the quality, cost, and reliability of its products and services. However, the supplier selection problem can become highly complex due to the uncertainties and vagueness associated with it. To overcome these complexities, multi-criteria decision analysis, and fuzzy logic have been used to incorporate uncertainties and vagueness into the supplier selection process. These techniques can help organizations make informed decisions and mitigate the risks associated with supplier selection. In this article, a complex picture fuzzy soft set (cpFSS), a generalized fuzzy set-like structure, is developed to deal with information-based uncertainties involved in the supplier selection process. It can maintain the expected information-based periodicity by introducing amplitude and phase terms. The amplitude term is meant for fuzzy membership, and the phase term is for managing its periodicity within the complex plane. The cpFSS also facilitates the decision-makers by allowing them the opportunity to provide their neutral grade-based opinions for objects under observation. Firstly, the essential notions and set-theoretic operations of cpFSS are investigated and illustrated with examples. Secondly, a MADM-based algorithm is proposed by describing new matrix-based aggregations of cpFSS like the core matrix, maximum and minimum decision value matrices, and score. Lastly, the proposed algorithm is implemented in real-world applications with the aim of selecting a suitable supplier for the provision of required materials for construction projects. With the sensitivity analysis of score values through Pythagorean means, it can be concluded that the results and rankings of the suppliers are consistent. Moreover, through structural comparison, the proposed structure is proven to be more flexible and reliable as compared to existing fuzzy set-like structures.

# INTRODUCTION

For every organization, choosing a supplier is a crucial decision-making process because it has an immediate impact on the reliability, pricing, and quality of their goods and services. The supplier selection problem (SSP), however, can become extremely complicated because of the ambiguities and uncertainties that surround it. After going through the literature (*Saputro, Figueira & Almada-Lobo, 2022*; *De Boer, Labro & Morlacchi, 2001*; *Altinoz, Kilduff & Winchester Jr, 2001*; *Bhutta, 2003*; *Jain, Benyoucef & Deshmukh, 2009*; *Amorim et al., 2016*), some of the complexities are:

1. Uncertainty in supplier performance: Given how challenging it is to precisely anticipate suppliers' future performance, there is always some degree of uncertainty around their performance as the chosen provider. The success of the provider may be impacted by a number of variables, including market trends, natural disasters, labour strikes, *etc*. The supplier selection problem may become very complex as a result of these uncertainties.

2. Vagueness in decision criteria: The selection criteria for suppliers are sometimes nebulous and imprecise. For instance, several organizations may interpret the criterion "quality" in different ways. The supplier selection process may become extremely complex as a result of ambiguity, which may give rise to various interpretations and judgments.

3. Incomplete information: The choice of a provider may become more difficult if there is little or incomplete information available about them. The supplier's financial soundness, production capability, and quality management procedures may not be fully disclosed to organizations. As it is challenging to make an informed judgment based on insufficient information, this missing information may make the supplier selection process extremely complex.

4. Multiple criteria: Typically, the SSP takes into account a number of factors, including price, quality, delivery time, and supplier reputation. Prioritizing these criteria might be difficult because they may contradict one another. Furthermore, the weights given to each criterion can change depending on the requirements of the organization, making the supplier selection problem extremely complex.

Organizations can utilize decision-making techniques like multi-criteria decision analysis, and fuzzy logic to include uncertainty and ambiguity in the SSP in order to get around these challenges. These methods can aid businesses in decision-making and risk mitigation related to supplier selection. Dealing with informational uncertainties has been a challenging problem for researchers. Several algebraic models have already been introduced to cope with such informational ambiguities. Picture fuzzy set (pFS) (*Cuong & Kreinovich, 2013*) has much significance in this regard. It is an extension of Zadeh's fuzzy set (ZFS) Zadeh1, Atanassov's intuitionistic fuzzy set (AIFS) (*Atanassov, 1986*) and refined pythagorean fuzzy set (*Saeed, Ahmad & Rahman, 2023*). *Saha, Reddy & Kumar (2022)* discussed the classification problem based on the formulations of fuzzy similarity measures and Archimedean-Dombi aggregation operator. Since the ZFS measures the belongingness of an element under observation by its relevant membership grade between 0 and 1 and the AIFS computes such feature by its related membership and non membership

grades that both take values from [0,1] provided that the sum of both membership and non membership grades must lie in [0,1]. But in pFS, the belongingness of an element in a universal set is characterized by three dimensional function which assigns positive membership grade, neutral membership grade and negative membership grade within [0,1] such that the sum of these three components must be contained in [0,1]. In fact, the pFS is introduced to tackle the situations where human opinions involve multiple options like no, refusal, yes. This feature has increased the flexibility of pFS. The idea of pFS has attracted the attention of many researchers like *Cuong & Pham (2015)* introduced some fuzzy logic operators for pFS. *Singh (2015)* and *Ganie, Singh & Bhatia (2020)* computed correlation coefficient for pFS with discussion on their properties and applications. *Wei (2018)* and *Wei & Gao (2018)* formulated similarity measures for pFS with their implementation in various fields. As information-based periodicity is often observed in raw data embedded with uncertainties. To cope with such situation, the idea of pFS has been extended to complex picture fuzzy set (cpFS) (*Akram, Bashir & Garg, 2020a*; *Qu et al., 2022*) which is the generalization of complex fuzzy set (cFS) (*Ramot et al., 2002*) and complex intuitionistic fuzzy set (cIFS) (*Alkouri & Salleh, 2012*). In cpFS, the codomain of a three dimensional function is a unit circle in the complex plane which is characterized by two terms: amplitude term and phase term. First term denotes the membership value and the second term represents its periodicity. *Yazdanbakhsh & Dick (2018)* and *Tamir, Rishe & Kandel (2015)* made systematic reviews on cFS logic. *Hu et al. (2018)* formulated the distance between cFSs and discussed the continuity of cFS operations. *Garg & Rani (2019)* and *Garg & Rani (2020)* discussed the various information measures, aggregation operations and ranking techniques for cIFSs. *Akram, Bashir & Garg (2020b)* and *Liu, Akram & Bashir (2021)* investigated some aggregation operators of cpFSs and applied them in decision making problems. The above-mentioned fuzzy set-like models are not compatible with parameterization mode; therefore, to make adequate with such setting, *Molodtsov (1999)* introduced the idea of soft set (SST). Thus, the theories of FS, IFS, pFS, cFS, cIFS and cpFS are extended for soft set to develop novel structures fuzzy soft set (FSS) (*Maji, Biswas & Roy, 2001a*), intuitionistic fuzzy soft set (IFSS) (*Maji, Biswas & Roy, 2001b*), picture fuzzy soft set (pFSS) (*Cuong & Kreinovich, 2014*), complex fuzzy soft set (cFSS) (*Thirunavukarasu, Suresh & Ashokkumar, 2017*), complex intuitionistic fuzzy soft set (cIFSS) (*Kumar & Bajaj, 2014*; *Ali et al., 2021*) and complex picture fuzzy soft set (cpFSS) (*Shanthi, Umamakeswari & Saranya, 2022*) respectively. *Rahman et al. (2020)*, *Ihsan et al. (2021)* and *Ihsan, Saeed & Rahman (2021)* made rich contributions refinement in fuzzy set-like structures and convexity in SST-like environments.

## Relevant literature

Supply chain management (SCMG) is the management of the stream of merchandise and services between producers and sites. This can contain the shifting and collection of unprocessed stuff, under-process work evaluation, prepared materials, and the order placement formalities from the production point to the utilization point. Due to the involvement of various criteria, SCMG has been considered as a multi-criteria decision-making (MCDM) problem with the association of uncertainties by the researchers (*Erceg*

*& Mularifović, 2019*; *Tamošaitiene et al., 2017*). *Chen & Su (2022)* used blockchain plus to optimize trust propagation on SCMG. *Xiao, Chen & Li (2012)* employed an innovative approach to discuss SSP by integrating the ideas of FCM and FSS with the consideration of risk factors. *Zulqarnain et al. (2021a)* and *Zulqarnain et al. (2021b)* introduced the notions of aggregation average operators with ordered weights under Pythagorean FSS and validated the notions by applying them in SSP. *Chang (2019)* made a discussion on SSP based on aggregation average operators with weights under IFSS. *Chatterjee, Mukherjee & Kar (2018)* used the ideas of linguistic terms, rough sets, and fuzzy numbers for approximating the various aspects of SSP. *Wang, Cheng & Huang (2009)* and *Junior, Osiro & Carpinetti (2014)* discussed SSP by adopting the fuzzy AHP and fuzzy TOPSIS techniques. *Büyüközkan & Çifçi (2011)* and *Büyüközkan & Göçer (2017)* argued the sustainability of SSP by considering incomplete information and axiomatic design respectively. *Patra & Mondal (2016)* integrated the techniques of FS risk analysis and the balanced solution to study SSP. *Agarwal, Biswas & Hanmandlu (2013)* generalized IFSS with the entitlement of the moderator's opinion about the original evaluation and applied the generalized structure to SSP. *García et al. (2013)* designed a robust decision support method for the evaluation of suppliers in SSP by considering Kraljic's terminology, *i.e.*, basic products. *Thao (2021)* used Archimedean t-conorms to formulate divergence measures and entropies for IFS and validated the idea by applying it in SSP. Similarly, *Zhao et al. (2017)* and *Khaleie, Fasanghari & Tavassoli (2012)* discussed SSP by adopting MCDM techniques under the IFS environment. *Liu & Wang (2022)* generalized the idea of FSS and applied it in SSP.

## Research motivation

With the keen observation of contributions provided in the above-mentioned literature, it is clear that the following important features of decision-making have been ignored:

1. While dealing with large amounts of data, the periodicity of information is often encountered, which may greatly affect the decision. This kind of issue is managed by using complex settings that tackle periodicity with the help of amplitude and phase terms.
2. Sometimes the decision-makers intend to be neutral while approximating any alternative based on suitable attributes. This issue is usually resolved with the entitlement of a suitable neutral grade, which increases the flexibility and reliability of the decision system.
3. Computational complexities lead to a decrease in the readability and understanding of any concept, therefore, it is pertinent to use easy and understandable computations to attract and facilitate the multidisciplinary community.

In the present study, all the above-stated issues are addressed by characterizing the idea of cpFSS. Its complex part (amplitude and phase terms) is meant to manage the first issue, and the picture fuzzy setting is meant to capture the second issue. Similarly, the third issue is addressed by using easy computations based on matrix manipulations.

## Salient contributions

The main objectives of this study are outlined as:

1. The concept of cpFSS discussed by *Shanthi, Umamakeswari & Saranya (2022)* is reviewed and modified with the characterization of some new properties and aggregation operations.

2. By considering the nature of SCMG as an uncertain MCDM problem, a robust decision support system is designed which is assisted by the proposal of an algorithm based on the aggregation operations of cpFSSs for selecting appropriate suppliers for construction projects.

The remaining article is systematized as some essential terms are recalled in 'Preliminaries', elementary notions of cpFSS and its aggregation operations are conceptualized in 'Characterization of elementary notions of CPFSSS', an MCDM based decision support system is constructed in 'Application of CPFSS in decision-making' and the article is concluded in the last section.

## PRELIMINARIES

This section recalls some definitions which are necessary for understanding the main concept. The symbols $\hat{\mathbb{S}}$, $\mathbb{I}^{[0,1]}$, $j$ and $2^{\hat{\mathbb{S}}}$ denote the initial space of objects, closed unit interval, $\sqrt{-1}$ and power set of $\hat{\mathbb{S}}$.

**Definition 2.1** *(Zadeh, 1965) If $\hat{M}_{\hat{T}}(\hat{s})$ is the truth membership grade of $\hat{s} \in \hat{\mathbb{S}}$ then FS $\hat{A}_F$ is defined as $\hat{A}_F = \left\{ (\hat{s}, \hat{M}_{\hat{T}}(\hat{s})) : \hat{s} \in \hat{\mathbb{S}} \right\}$ where $\hat{M}_{\hat{T}} : \hat{\mathbb{S}} \to \mathbb{I}^{[0,1]}$ is membership mapping provided that $0 \leq \hat{M}_{\hat{T}}(\hat{s}) \leq 1$. The family of all fuzzy subsets is represented by $\Sigma_{FS}$.*

**Definition 2.2** *(Atanassov, 1986) If $\hat{M}_{\hat{T}}(\hat{s})$ and $\hat{M}_{\hat{F}}(\hat{s})$ are truth and false membership grades of $\hat{s} \in \hat{\mathbb{S}}$ then IFS $\hat{A}_{IF}$ is defined as $\hat{A}_{IF} = \left\{ (\hat{s}, \langle \hat{M}_{\hat{T}}(\hat{s}), \hat{M}_{\hat{F}}(\hat{s}) \rangle) : \hat{s} \in \hat{\mathbb{S}} \right\}$ where $\hat{M}_{\hat{T}}, \hat{M}_{\hat{F}} : \hat{\mathbb{S}} \to \mathbb{I}^{[0,1]}$ are membership mappings provided that $0 \leq \hat{M}_{\hat{T}}(\hat{s}) + \hat{M}_{\hat{F}}(\hat{s}) \leq 1$ with hesitancy grade $\hat{M}_{\hat{H}}(\hat{s}) = 1 - (\hat{M}_{\hat{T}}(\hat{s}) + \hat{M}_{\hat{F}}(\hat{s}))$ within $\mathbb{I}^{[0,1]}$. The family of all IF-subsets is represented by $\Sigma_{IFS}$.*

**Definition 2.3** *(Cuong & Kreinovich, 2013) If $\hat{M}_{\hat{P}}(\hat{s})$, $\hat{M}_{\hat{N}e}(\hat{s})$ and $\hat{M}_{\hat{N}}(\hat{s})$ are positive, neutral and negative membership grades of $\hat{s} \in \hat{\mathbb{S}}$ then pFS $\hat{\Omega}$ is defined as $\hat{\Omega} = \left\{ (\hat{s}, \langle \hat{M}_{\hat{P}}(\hat{s}), \hat{M}_{\hat{N}e}(\hat{s}), \hat{M}_{\hat{N}}(\hat{s}) \rangle) : \hat{s} \in \hat{\mathbb{S}} \right\}$ where $\hat{M}_{\hat{P}}, \hat{M}_{\hat{N}e}, \hat{M}_{\hat{N}} : \hat{\mathbb{S}} \to \mathbb{I}^{[0,1]}$ are membership mappings provided that $0 \leq \hat{M}_{\hat{P}}(\hat{s}) + \hat{M}_{\hat{N}e}(\hat{s}) + \hat{M}_{\hat{N}}(\hat{s}) \leq 1$ with refusal membership grade $\hat{M}_{\hat{R}}(\hat{s}) = 1 - (\hat{M}_{\hat{P}}(\hat{s}) + \hat{M}_{\hat{N}e}(\hat{s}) + \hat{M}_{\hat{N}}(\hat{s}))$ within $\mathbb{I}^{[0,1]}$.*

**Definition 2.4** *(Akram, Bashir & Garg, 2020a; Qu et al., 2022) If $\hat{M}_{\hat{P}}(\hat{s})$, $\hat{M}_{\hat{N}e}(\hat{s})$ and $\hat{M}_{\hat{N}}(\hat{s})$ are positive, neutral and negative membership grades of $\hat{s} \in \hat{\mathbb{S}}$ then cpFS $\hat{\Theta}$ is defined as $\hat{\Theta} = \left\{ (\hat{s}, \langle \hat{M}_{\hat{P}}(\hat{s}), \hat{M}_{\hat{N}e}(\hat{s}), \hat{M}_{\hat{N}}(\hat{s}) \rangle) : \hat{s} \in \hat{\mathbb{S}} \right\}$ where $\hat{M}_{\hat{P}}, \hat{M}_{\hat{N}e}, \hat{M}_{\hat{N}} : \hat{\mathbb{S}} \to \mathbb{C}^{[0,1]}$ are complex fuzzy membership mappings such that $\hat{M}_{\hat{P}}(\hat{s}) = \alpha_{\hat{P}}(\hat{s}) \exp^{j\beta_{\hat{P}}(\hat{s})}$, $\hat{M}_{\hat{N}e}(\hat{s}) = \alpha_{\hat{N}e}(\hat{s}) \exp^{j\beta_{\hat{N}e}(\hat{s})}$ and $\hat{M}_{\hat{N}}(\hat{s}) = \alpha_{\hat{N}}(\hat{s}) \exp^{j\beta_{\hat{N}}(\hat{s})}$ provided that $0 \leq \alpha_{\hat{P}}(\hat{s}) + \alpha_{\hat{N}e}(\hat{s}) + \alpha_{\hat{N}}(\hat{s}) \leq 1$ and $0 \leq \beta_{\hat{P}}(\hat{s}) + \beta_{\hat{N}e}(\hat{s}) + \beta_{\hat{N}}(\hat{s}) \leq 2\pi$. The $\alpha_{\hat{P}}(\hat{s}), \alpha_{\hat{N}e}(\hat{s})$, $\alpha_{\hat{N}}(\hat{s})$ are called the amplitude terms and $\beta_{\hat{P}}(\hat{s}), \beta_{\hat{N}e}(\hat{s})$, and $\beta_{\hat{N}}(\hat{s})$ are known as phase terms. The refusal membership grade $\hat{M}_{\hat{R}}(\hat{s}) = [1 - \alpha_{\hat{P}}(\hat{s}) - \alpha_{\hat{N}e}(\hat{s}) - \alpha_{\hat{N}}(\hat{s})] \exp^{j[2\pi - \beta_{\hat{P}}(\hat{s}) - \beta_{\hat{N}e}(\hat{s}) - \beta_{\hat{N}}(\hat{s})]}$ within $\mathbb{C}^{[0,1]}$. The family of all cpFSs over $\hat{\mathbb{S}}$ is symbolized as $\Delta_{cpFS}(\hat{\mathbb{S}})$.*

**Definition 2.5** *(Molodtsov, 1999) If $\hat{\Xi}$ is consisting of attributes then an SST $\hat{\Psi}$ is defined as $\hat{\Psi} = \left\{ \left( \hat{a}, \hat{\zeta}(\hat{a}) \right) : \hat{a} \in \hat{\Xi} \right\}$ where $\hat{\zeta} : \hat{\Xi} \to 2^{\hat{\mathbb{S}}}$ is an approximate mapping with $\hat{\zeta}(\hat{a})$ as $\hat{a}$-approximate element in $\hat{\mathbb{S}}$.*

**Definition 2.6** *(Maji, Biswas & Roy, 2001a) If $\hat{\Xi}$ is consisting of attributes then FSS $\hat{\Psi}_F$ is defined as $\hat{\Psi}_F = \left\{ \left( \hat{a}, \hat{\zeta}_F(\hat{a}) \right) : \hat{a} \in \hat{\Xi} \right\}$ where $\hat{\zeta}_F : \hat{\Xi} \to \Sigma_{FS}$ is an approximate mapping with $\hat{\zeta}_F(\hat{a})$ as $\hat{a}$-approximate element in $\hat{\mathbb{S}}$.*

**Definition 2.7** *(Maji, Biswas & Roy, 2001b) If $\hat{\Xi}$ is consisting of attributes then an IFSS $\hat{\Psi}_{IF}$ is defined as $\hat{\Psi}_{IF} = \left\{ \left( \hat{a}, \hat{\zeta}_{IF}(\hat{a}) \right) : \hat{a} \in \hat{\Xi} \right\}$ where $\hat{\zeta}_{IF} : \hat{\Xi} \to \Sigma_{IF}$ is an approximate mapping with $\hat{\zeta}_{IF}(\hat{a})$ as $\hat{a}$-approximate element in $\hat{\mathbb{S}}$.*

## CHARACTERIZATION OF ELEMENTARY NOTIONS OF CPFSSS

This section is aimed to investigate some elementary notions and operations of cpFSSs after modifying the concepts provided by *Shanthi, Umamakeswari & Saranya (2022)*.

**Definition 3.1** If $\hat{\Lambda}$ is the subset of a set $\hat{\Xi}$ consisting of attributes and $\hat{M}_{\hat{P}}(\hat{a})$, $\hat{M}_{\hat{Ne}}(\hat{a})$, $\hat{M}_{\hat{N}}(\hat{a})$ are positive, neutral and negative membership grades of $\hat{a} \in \hat{\Lambda}$ then the cpFSS $\hat{\Pi}$ is defined as where $\hat{M}_{\hat{P}}, \hat{M}_{\hat{Ne}}, \hat{M}_{\hat{N}} : \hat{\Lambda} \to \Delta_{cpFS}(\hat{\mathbb{S}})$ are complex fuzzy approximate mappings such that $\hat{M}_{\hat{P}}(\hat{a}) = \alpha_{\hat{P}}(\hat{a}) \exp^{j\beta_{\hat{P}}(\hat{a})}$, $\hat{M}_{\hat{Ne}}(\hat{a}) = \alpha_{\hat{Ne}}(\hat{a}) \exp^{j\beta_{\hat{Ne}}(\hat{a})}$ and $\hat{M}_{\hat{N}}(\hat{a}) = \alpha_{\hat{N}}(\hat{a}) \exp^{j\beta_{\hat{N}}(\hat{a})}$ provided that $0 \leq \alpha_{\hat{P}}(\hat{a}) + \alpha_{\hat{Ne}}(\hat{a}) + \alpha_{\hat{N}}(\hat{a}) \leq 1$ and $0 \leq \beta_{\hat{P}}(\hat{a}) + \beta_{\hat{Ne}}(\hat{a}) + \beta_{\hat{N}}(\hat{a}) \leq 2\pi$. The $\alpha_{\hat{P}}(\hat{a}), \alpha_{\hat{Ne}}(\hat{a})$, $\alpha_{\hat{N}}(\hat{a})$ are called the amplitude terms and $\beta_{\hat{P}}(\hat{a}), \beta_{\hat{Ne}}(\hat{a})$, and $\beta_{\hat{N}}(\hat{a})$ are known as phase terms. The refusal membership grade

$$\hat{M}_{\hat{R}}(\hat{a}) = [1 - \alpha_{\hat{P}}(\hat{a}) - \alpha_{\hat{Ne}}(\hat{a}) - \alpha_{\hat{N}}(\hat{a})] \exp^{j[2\pi - \beta_{\hat{P}}(\hat{a}) - \beta_{\hat{Ne}}(\hat{a}) - \beta_{\hat{N}}(\hat{a})]}$$

within $\mathbb{C}^{[0,1]}$. For convenience, $\langle \alpha_{\hat{P}}(\hat{a}) \exp^{j\beta_{\hat{P}}(\hat{a})}, \alpha_{\hat{Ne}}(\hat{a}) \exp^{j\beta_{\hat{Ne}}(\hat{a})}, \alpha_{\hat{N}}(\hat{a}) \exp^{j\beta_{\hat{N}}(\hat{a})} \rangle$ is called a complex picture fuzzy soft number (cpFSN). The family of all cpFSSs over $\hat{\mathbb{S}}$ is symbolized as $\Delta_{cpFSS}(\hat{\mathbb{S}})$.

**Example 3.2** Let $\hat{\mathbb{S}} = \{\hat{s}_1, \hat{s}_2, \hat{s}_3, \hat{s}_4\}$ be an initial space of objects and $\hat{\Xi} = \{\hat{a}_1, \hat{a}_2, \hat{a}_3, \hat{a}_4, \hat{a}_5, \hat{a}_6\}$ be a set of attributes with $\hat{\Lambda} = \{\hat{a}_1, \hat{a}_2, \hat{a}_5, \hat{a}_6\} \subseteq \hat{\Xi}$ then approximate elements of cpFSS $\hat{\Pi}$ are computed as

$$\delta(\hat{a}_1) = \begin{Bmatrix} \left( \hat{s}_1, \langle 0.11 \exp^{j2\pi(0.22)}, 0.25 \exp^{j2\pi(0.31)}, 0.42 \exp^{j2\pi(0.15)} \rangle \right), \\ \left( \hat{s}_2, \langle 0.23 \exp^{j2\pi(0.18)}, 0.32 \exp^{j2\pi(0.21)}, 0.51 \exp^{j2\pi(0.22)} \rangle \right), \\ \left( \hat{s}_3, \langle 0.33 \exp^{j2\pi(0.25)}, 0.35 \exp^{j2\pi(0.30)}, 0.20 \exp^{j2\pi(0.45)} \rangle \right), \\ \left( \hat{s}_4, \langle \rangle 0.15 \exp^{j2\pi(0.30)}, 0.21 \exp^{j2\pi(0.45)}, 0.33 \exp^{j2\pi(0.60)} \langle \rangle \right) \end{Bmatrix},$$

$$\delta(\hat{a}_2) = \begin{Bmatrix} \left( \hat{s}_1, \langle 0.12 \exp^{j2\pi(0.24)}, 0.26 \exp^{j2\pi(0.30)}, 0.35 \exp^{j2\pi(0.40)} \rangle \right), \\ \left( \hat{s}_2, \langle 0.22 \exp^{j2\pi(0.31)}, 0.25 \exp^{j2\pi(0.34)}, 0.28 \exp^{j2\pi(0.37)} \rangle \right), \\ \left( \hat{s}_3, \langle 0.23 \exp^{j2\pi(0.32)}, 0.26 \exp^{j2\pi(0.35)}, 0.29 \exp^{j2\pi(0.38)} \rangle \right), \\ \left( \hat{s}_4, \langle 0.24 \exp^{j2\pi(0.33)}, 0.27 \exp^{j2\pi(0.36)}, 0.30 \exp^{j2\pi(0.39)} \rangle \right) \end{Bmatrix},$$

$$\delta(\hat{a}_5) = \begin{Bmatrix} (\hat{s}_1, \langle 0.11\,exp^{j2\pi(0.41)}, 0.12\,exp^{j2\pi(0.31)}, 0.40\,exp^{j2\pi(0.51)} \rangle), \\ (\hat{s}_2, \langle 0.21\,exp^{j2\pi(0.30)}, 0.24\,exp^{j2\pi(0.33)}, 0.27\,exp^{j2\pi(0.36)} \rangle), \\ (\hat{s}_3, \langle 0.22\,exp^{j2\pi(0.31)}, 0.25\,exp^{j2\pi(0.34)}, 0.28\,exp^{j2\pi(0.37)} \rangle), \\ (\hat{s}_4, \langle 0.23\,exp^{j2\pi(0.32)}, 0.26\,exp^{j2\pi(0.35)}, 0.29\,exp^{j2\pi(0.38)} \rangle) \end{Bmatrix},$$

$$\delta(\hat{a}_6) = \begin{Bmatrix} (\hat{s}_1, \langle 0.13\,exp^{j2\pi(0.44)}, 0.14\,exp^{j2\pi(0.55)}, 0.44\,exp^{j2\pi(0.66)} \rangle), \\ (\hat{s}_2, \langle 0.25\,exp^{j2\pi(0.18)}, 0.28\,exp^{j2\pi(0.21)}, 0.31\,exp^{j2\pi(0.24)} \rangle), \\ (\hat{s}_3, \langle 0.26\,exp^{j2\pi(0.19)}, 0.29\,exp^{j2\pi(0.22)}, 0.32\,exp^{j2\pi(0.25)} \rangle), \\ (\hat{s}_4, \langle 0.27\,exp^{j2\pi(0.20)}, 0.30\,exp^{j2\pi(0.23)}, 0.33\,exp^{j2\pi(0.26)} \rangle) \end{Bmatrix}.$$

*The cpFSS $\hat{\Pi}$ is constructed as $\hat{\Pi} = \{(\hat{a}_1, \hat{\delta}(\hat{a}_1)), (\hat{a}_2, \hat{\delta}(\hat{a}_2)), (\hat{a}_5, \hat{\delta}(\hat{a}_5)), (\hat{a}_6, \hat{\delta}(\hat{a}_6))\}$. It can be represented in matrix notation as $\hat{\Pi} =$*

$$\begin{pmatrix} \langle 0.11^{0.22}, 0.25^{0.31}, 0.42^{0.15} \rangle & \langle 0.23^{0.18}, 0.32^{0.21}, 0.51^{0.22} \rangle & \langle 0.33^{0.25}, 0.35^{0.30}, 0.20^{0.45} \rangle & \langle 0.15^{0.30}, 0.21^{0.45}, 0.33^{0.60} \rangle \\ \langle 0.12^{0.24}, 0.26^{0.30}, 0.35^{0.40} \rangle & \langle 0.22^{0.31}, 0.25^{0.34}, 0.28^{0.37} \rangle & \langle 0.23^{0.32}, 0.26^{0.35}, 0.29^{0.38} \rangle & \langle 0.24^{0.33}, 0.27^{0.36}, 0.30^{0.39} \rangle \\ \langle 0.11^{0.41}, 0.12^{0.31}, 0.40^{0.51} \rangle & \langle 0.21^{0.30}, 0.24^{0.33}, 0.27^{0.36} \rangle & \langle 0.22^{0.31}, 0.25^{0.34}, 0.28^{0.37} \rangle & \langle 0.23^{0.32}, 0.26^{0.35}, 0.29^{0.38} \rangle \\ \langle 0.13^{0.44}, 0.14^{0.55}, 0.44^{0.66} \rangle & \langle 0.25^{0.18}, 0.28^{0.21}, 0.31^{0.24} \rangle & \langle 0.26^{0.19}, 0.29^{0.22}, 0.32^{0.25} \rangle & \langle 0.27^{0.20}, 0.30^{0.23}, 0.33^{0.26} \rangle \end{pmatrix}.$$

**Definition 3.3** For $\hat{\Pi}_1, \hat{\Pi}_2 \in \Delta_{cpFSS}(\hat{\mathbb{S}})$ then following set theoretic operations are valid:

1. $\hat{\Pi}_1 \subseteq \hat{\Pi}_2$ if $\alpha^1_{\hat{P}}(\hat{a}) \leq \alpha^2_{\hat{P}}(\hat{a})$, $\alpha^1_{\hat{N}e}(\hat{a}) \geq \alpha^2_{\hat{N}e}(\hat{a})$, $\alpha^1_{\hat{N}}(\hat{a}) \geq \alpha^2_{\hat{N}}(\hat{a})$ and $\beta^1_{\hat{P}}(\hat{a}) \leq \beta^2_{\hat{P}}(\hat{a})$, $\beta^1_{\hat{N}e}(\hat{a}) \geq \beta^2_{\hat{N}e}(\hat{a})$, $\beta^1_{\hat{N}}(\hat{a}) \geq \beta^2_{\hat{N}}(\hat{a})$.

2. *The* $\hat{\Pi}_1 \cap \hat{\Pi}_2$ is another cpFSS $\hat{\Pi}_3$ with $\alpha^3_{\hat{P}}(\hat{a}) = min\{\alpha^1_{\hat{P}}(\hat{a}), \alpha^2_{\hat{P}}(\hat{a})\}$, $\alpha^3_{\hat{N}e}(\hat{a}) = max\{\alpha^1_{\hat{N}e}(\hat{a}), \alpha^2_{\hat{N}e}(\hat{a})\}$, $\alpha^3_{\hat{N}}(\hat{a}) = max\{\alpha^1_{\hat{N}}(\hat{a}), \alpha^2_{\hat{N}}(\hat{a})\}$ and $\beta^3_{\hat{P}}(\hat{a}) = min\{\beta^1_{\hat{P}}(\hat{a}), \beta^2_{\hat{P}}(\hat{a})\}$, $\beta^3_{\hat{N}e}(\hat{a}) = max\{\beta^1_{\hat{N}e}(\hat{a}), \beta^2_{\hat{N}e}(\hat{a})\}$, $\beta^3_{\hat{N}}(\hat{a}) = max\{\beta^1_{\hat{N}}(\hat{a}), \beta^2_{\hat{N}}(\hat{a})\}$.

3. The $\hat{\Pi}_1 \cup \hat{\Pi}_2$ is another cpFSS $\hat{\Pi}_4$ with $\alpha^4_{\hat{P}}(\hat{a}) = max\{\alpha^1_{\hat{P}}(\hat{a}), \alpha^2_{\hat{P}}(\hat{a})\}$, $\alpha^4_{\hat{N}e}(\hat{a}) = min\{\alpha^1_{\hat{N}e}(\hat{a}), \alpha^2_{\hat{N}e}(\hat{a})\}$, $\alpha^4_{\hat{N}}(\hat{a}) = min\{\alpha^1_{\hat{N}}(\hat{a}), \alpha^2_{\hat{N}}(\hat{a})\}$ and $\beta^4_{\hat{P}}(\hat{a}) = max\{\beta^1_{\hat{P}}(\hat{a}), \beta^2_{\hat{P}}(\hat{a})\}$, $\beta^4_{\hat{N}e}(\hat{a}) = min\{\beta^1_{\hat{N}e}(\hat{a}), \beta^2_{\hat{N}e}(\hat{a})\}$, $\beta^4_{\hat{N}}(\hat{a}) = min\{\beta^1_{\hat{N}}(\hat{a}), \beta^2_{\hat{N}}(\hat{a})\}$.

4. The $\hat{\Pi}^c_1$ is again cpFSS with $\alpha^c_{\hat{P}}(\hat{a}) = \alpha^1_{\hat{N}}(\hat{a})$, $\alpha^c_{\hat{N}e}(\hat{a}) = \alpha^1_{\hat{N}e}(\hat{a})$, $\alpha^c_{\hat{N}}(\hat{a}) = \alpha^1_{\hat{P}}(\hat{a})$ and $\beta^c_{\hat{P}}(\hat{a}) = \beta^1_{\hat{N}}(\hat{a})$, $\beta^c_{\hat{N}e}(\hat{a}) = \beta^1_{\hat{N}e}(\hat{a})$, $\beta^c_{\hat{N}}(\hat{a}) = \beta^1_{\hat{P}}(\hat{a})$.

**Example 3.4** Reassuming the data from Example 3.2, the following two cpFSSs are constructed with matrix representations as given below

$\hat{\Pi}_1 =$

$$\begin{pmatrix} \langle 0.10^{0.21}, 0.26^{0.32}, 0.43^{0.16} \rangle & \langle 0.22^{0.17}, 0.33^{0.22}, 0.52^{0.23} \rangle & \langle 0.32^{0.24}, 0.36^{0.31}, 0.21^{0.46} \rangle & \langle 0.14^{0.29}, 0.22^{0.46}, 0.34^{0.61} \rangle \\ \langle 0.11^{0.23}, 0.27^{0.31}, 0.36^{0.41} \rangle & \langle 0.21^{0.30}, 0.26^{0.35}, 0.29^{0.38} \rangle & \langle 0.22^{0.31}, 0.27^{0.36}, 0.30^{0.39} \rangle & \langle 0.23^{0.32}, 0.28^{0.37}, 0.31^{0.40} \rangle \\ \langle 0.10^{0.40}, 0.13^{0.32}, 0.41^{0.52} \rangle & \langle 0.20^{0.29}, 0.25^{0.34}, 0.28^{0.37} \rangle & \langle 0.21^{0.30}, 0.26^{0.35}, 0.29^{0.38} \rangle & \langle 0.22^{0.31}, 0.27^{0.36}, 0.30^{0.39} \rangle \\ \langle 0.12^{0.43}, 0.15^{0.56}, 0.45^{0.67} \rangle & \langle 0.24^{0.17}, 0.29^{0.22}, 0.32^{0.25} \rangle & \langle 0.25^{0.18}, 0.30^{0.23}, 0.33^{0.26} \rangle & \langle 0.26^{0.19}, 0.31^{0.24}, 0.34^{0.27} \rangle \end{pmatrix},$$

$\hat{\Pi}_2 =$

$$\begin{pmatrix} \langle 0.11^{0.22}, 0.25^{0.31}, 0.42^{0.15} \rangle & \langle 0.23^{0.18}, 0.32^{0.21}, 0.51^{0.22} \rangle & \langle 0.33^{0.25}, 0.35^{0.30}, 0.20^{0.45} \rangle & \langle 0.15^{0.30}, 0.21^{0.45}, 0.33^{0.60} \rangle \\ \langle 0.12^{0.24}, 0.26^{0.30}, 0.35^{0.40} \rangle & \langle 0.22^{0.31}, 0.25^{0.34}, 0.28^{0.37} \rangle & \langle 0.23^{0.32}, 0.26^{0.35}, 0.29^{0.38} \rangle & \langle 0.24^{0.33}, 0.27^{0.36}, 0.30^{0.39} \rangle \\ \langle 0.11^{0.41}, 0.12^{0.31}, 0.40^{0.51} \rangle & \langle 0.21^{0.30}, 0.24^{0.33}, 0.27^{0.36} \rangle & \langle 0.22^{0.31}, 0.25^{0.34}, 0.28^{0.37} \rangle & \langle 0.23^{0.32}, 0.26^{0.35}, 0.29^{0.38} \rangle \\ \langle 0.13^{0.44}, 0.14^{0.55}, 0.44^{0.66} \rangle & \langle 0.25^{0.18}, 0.28^{0.21}, 0.31^{0.24} \rangle & \langle 0.26^{0.19}, 0.29^{0.22}, 0.32^{0.25} \rangle & \langle 0.27^{0.20}, 0.30^{0.23}, 0.33^{0.26} \rangle \end{pmatrix}.$$

Then $\hat{\Pi}_1 \subseteq \hat{\Pi}_2$ and $\hat{\Pi}_1 \cap \hat{\Pi}_2 =$

$$\begin{pmatrix} \langle 0.10^{0.21}, 0.26^{0.32}, 0.43^{0.16} \rangle & \langle 0.22^{0.17}, 0.33^{0.22}, 0.52^{0.23} \rangle & \langle 0.32^{0.24}, 0.36^{0.31}, 0.21^{0.46} \rangle & \langle 0.14^{0.29}, 0.22^{0.46}, 0.34^{0.61} \rangle \\ \langle 0.11^{0.23}, 0.27^{0.31}, 0.36^{0.41} \rangle & \langle 0.21^{0.30}, 0.26^{0.35}, 0.29^{0.38} \rangle & \langle 0.22^{0.31}, 0.27^{0.36}, 0.30^{0.39} \rangle & \langle 0.23^{0.32}, 0.28^{0.37}, 0.31^{0.40} \rangle \\ \langle 0.10^{0.40}, 0.13^{0.32}, 0.41^{0.52} \rangle & \langle 0.20^{0.29}, 0.25^{0.34}, 0.28^{0.37} \rangle & \langle 0.21^{0.30}, 0.26^{0.35}, 0.29^{0.38} \rangle & \langle 0.22^{0.31}, 0.27^{0.36}, 0.30^{0.39} \rangle \\ \langle 0.12^{0.43}, 0.15^{0.56}, 0.45^{0.67} \rangle & \langle 0.24^{0.17}, 0.29^{0.22}, 0.32^{0.25} \rangle & \langle 0.25^{0.18}, 0.30^{0.23}, 0.33^{0.26} \rangle & \langle 0.26^{0.19}, 0.31^{0.24}, 0.34^{0.27} \rangle \end{pmatrix}.$$

Similarly $\hat{\Pi}_1 \cup \hat{\Pi}_2 =$

$$\begin{pmatrix}
\langle 0.11^{0.22}, 0.25^{0.31}, 0.42^{0.15}\rangle & \langle 0.23^{0.18}, 0.32^{0.21}, 0.51^{0.22}\rangle & \langle 0.33^{0.25}, 0.35^{0.30}, 0.20^{0.45}\rangle & \langle 0.15^{0.30}, 0.21^{0.45}, 0.33^{0.60}\rangle \\
\langle 0.12^{0.24}, 0.26^{0.30}, 0.35^{0.40}\rangle & \langle 0.22^{0.31}, 0.25^{0.34}, 0.28^{0.37}\rangle & \langle 0.23^{0.32}, 0.26^{0.35}, 0.29^{0.38}\rangle & \langle 0.24^{0.33}, 0.27^{0.36}, 0.30^{0.39}\rangle \\
\langle 0.11^{0.41}, 0.12^{0.31}, 0.40^{0.51}\rangle & \langle 0.21^{0.30}, 0.24^{0.33}, 0.27^{0.36}\rangle & \langle 0.22^{0.31}, 0.25^{0.34}, 0.28^{0.37}\rangle & \langle 0.23^{0.32}, 0.26^{0.35}, 0.29^{0.38}\rangle \\
\langle 0.13^{0.44}, 0.14^{0.55}, 0.44^{0.66}\rangle & \langle 0.25^{0.18}, 0.28^{0.21}, 0.31^{0.24}\rangle & \langle 0.26^{0.19}, 0.29^{0.22}, 0.32^{0.25}\rangle & \langle 0.27^{0.20}, 0.30^{0.23}, 0.33^{0.26}\rangle
\end{pmatrix},$$

and $\hat{\Pi}^c =$

$$\begin{pmatrix}
\langle 0.42^{0.15}, 0.25^{0.31}, 0.11^{0.22}\rangle & \langle 0.51^{0.22}, 0.32^{0.21}, 0.23^{0.18}\rangle & \langle 0.20^{0.45}, 0.35^{0.30}, 0.33^{0.25}\rangle & \langle 0.33^{0.60}, 0.21^{0.45}, 0.15^{0.30}\rangle \\
\langle 0.35^{0.40}, 0.26^{0.30}, 0.12^{0.24}\rangle & \langle 0.28^{0.37}, 0.25^{0.34}, 0.22^{0.31}\rangle & \langle 0.29^{0.38}, 0.26^{0.35}, 0.23^{0.32}\rangle & \langle 0.30^{0.39}, 0.27^{0.36}, 0.24^{0.33}\rangle \\
\langle 0.40^{0.51}, 0.12^{0.31}, 0.11^{0.41}\rangle & \langle 0.27^{0.36}, 0.24^{0.33}, 0.21^{0.30}\rangle & \langle 0.28^{0.37}, 0.25^{0.34}, 0.22^{0.31}\rangle & \langle 0.29^{0.38}, 0.26^{0.35}, 0.23^{0.32}\rangle \\
\langle 0.44^{0.66}, 0.14^{0.55}, 0.13^{0.44}\rangle & \langle 0.31^{0.24}, 0.28^{0.21}, 0.25^{0.18}\rangle & \langle 0.32^{0.25}, 0.29^{0.22}, 0.26^{0.19}\rangle & \langle 0.33^{0.26}, 0.30^{0.23}, 0.27^{0.20}\rangle
\end{pmatrix}.$$

# APPLICATION OF CPFSS IN DECISION-MAKING

This section is aimed to propose an MADM-based algorithm and then explained by a daily life problem of supplier selection based on aggregations of cpFSSs.

**Algorithm 1.** This algorithm consists of the following steps:

1. Considering the essential sets and opinions of decision makers, construct cpFSS $\hat{\Pi}$, *i.e.*, $\hat{\Pi} = \left\{ \left( \hat{a}, \langle \alpha_{\hat{P}}(\hat{a}) \exp^{j\beta_{\hat{P}}(\hat{a})}, \alpha_{\dot{N}e}(\hat{a}) \exp^{j\beta_{\dot{N}e}(\hat{a})}, \alpha_{\dot{N}}(\hat{a}) \exp^{j\beta_{\dot{N}}(\hat{a})} \rangle \right) : \hat{a} \in \hat{\Lambda} \right\}$.

2. Represent the cpFSS $\hat{\Pi}$ in matrix notation $\hat{\mho}_{p \times q}, p, q \in \mathbb{N}$, where $p$ and $q$ are the cardinalities of set of attributes and initial space of objects respectively.

$$\hat{\mho}_{p \times q} = \begin{pmatrix}
\hat{\omega}_{11} & \hat{\omega}_{12} & \cdots & \hat{\omega}_{1q} \\
\hat{\omega}_{21} & \hat{\omega}_{22} & \cdots & \hat{\omega}_{2q} \\
\hat{\omega}_{31} & \hat{\omega}_{32} & \cdots & \hat{\omega}_{3q} \\
\vdots & \vdots & \ddots & \vdots \\
\hat{\omega}_{p1} & \hat{\omega}_{p2} & \cdots & \hat{\omega}_{pq}
\end{pmatrix}$$

where

$\hat{\omega}_{11} = \langle \alpha_{\hat{P}}(\hat{a}_1)(\hat{s}_1) \exp^{j\beta_{\hat{P}}(\hat{a}_1)(\hat{s}_1)}, \alpha_{\dot{N}e}(\hat{a}_1)(\hat{s}_1) \exp^{j\beta_{\dot{N}e}(\hat{a}_1)(\hat{s}_1)}, \alpha_{\dot{N}}(\hat{a}_1)(\hat{s}_1) \exp^{j\beta_{\dot{N}}(\hat{a}_1)(\hat{s}_1)} \rangle$

$\hat{\omega}_{12} = \langle \alpha_{\hat{P}}(\hat{a}_1)(\hat{s}_2) \exp^{j\beta_{\hat{P}}(\hat{a}_1)(\hat{s}_2)}, \alpha_{\dot{N}e}(\hat{a}_1)(\hat{s}_2) \exp^{j\beta_{\dot{N}e}(\hat{a}_1)(\hat{s}_2)}, \alpha_{\dot{N}}(\hat{a}_1)(\hat{s}_2) \exp^{j\beta_{\dot{N}}(\hat{a}_1)(\hat{s}_2)} \rangle$

......................................................................................................................

......................................................................................................................

......................................................................................................................

$\hat{\omega}_{1q} = \langle \alpha_{\hat{P}}(\hat{a}_1)(\hat{s}_q) \exp^{j\beta_{\hat{P}}(\hat{a}_1)(\hat{s}_q)}, \alpha_{\dot{N}e}(\hat{a}_1)(\hat{s}_q) \exp^{j\beta_{\dot{N}e}(\hat{a}_1)(\hat{s}_q)}, \alpha_{\dot{N}}(\hat{a}_1)(\hat{s}_q) \exp^{j\beta_{\dot{N}}(\hat{a}_1)(\hat{s}_q)} \rangle$

$\hat{\omega}_{21} = \langle \alpha_{\hat{P}}(\hat{a}_2)(\hat{s}_1) \exp^{j\beta_{\hat{P}}(\hat{a}_2)(\hat{s}_1)}, \alpha_{\dot{N}e}(\hat{a}_2)(\hat{s}_1) \exp^{j\beta_{\dot{N}e}(\hat{a}_2)(\hat{s}_1)}, \alpha_{\dot{N}}(\hat{a}_2)(\hat{s}_1) \exp^{j\beta_{\dot{N}}(\hat{a}_2)(\hat{s}_1)} \rangle$

$\hat{\omega}_{22} = \langle \alpha_{\hat{P}}(\hat{a}_2)(\hat{s}_2) \exp^{j\beta_{\hat{P}}(\hat{a}_2)(\hat{s}_2)}, \alpha_{\dot{N}e}(\hat{a}_2)(\hat{s}_2) \exp^{j\beta_{\dot{N}e}(\hat{a}_2)(\hat{s}_2)}, \alpha_{\dot{N}}(\hat{a}_2)(\hat{s}_2) \exp^{j\beta_{\dot{N}}(\hat{a}_2)(\hat{s}_2)} \rangle$

......................................................................................................................

......................................................................................................................

......................................................................................................................

$\hat{\omega}_{2q} = \langle \alpha_{\hat{P}}(\hat{a}_2)(\hat{s}_q) \exp^{j\beta_{\hat{P}}(\hat{a}_2)(\hat{s}_q)}, \alpha_{\dot{N}e}(\hat{a}_2)(\hat{s}_q) \exp^{j\beta_{\dot{N}e}(\hat{a}_2)(\hat{s}_q)}, \alpha_{\dot{N}}(\hat{a}_2)(\hat{s}_q) \exp^{j\beta_{\dot{N}}(\hat{a}_2)(\hat{s}_q)} \rangle$

$\hat{\omega}_{31} = \langle \alpha_{\hat{P}}(\hat{a}_3)(\hat{s}_1) \exp^{j\beta_{\hat{P}}(\hat{a}_3)(\hat{s}_1)}, \alpha_{\dot{N}e}(\hat{a}_3)(\hat{s}_1) \exp^{j\beta_{\dot{N}e}(\hat{a}_3)(\hat{s}_1)}, \alpha_{\dot{N}}(\hat{a}_3)(\hat{s}_1) \exp^{j\beta_{\dot{N}}(\hat{a}_3)(\hat{s}_1)} \rangle$

$\hat{\omega}_{32} = \langle \alpha_{\hat{P}}(\hat{a}_3)(\hat{s}_2) \exp^{j\beta_{\hat{P}}(\hat{a}_3)(\hat{s}_2)}, \alpha_{\dot{N}e}(\hat{a}_3)(\hat{s}_2) \exp^{j\beta_{\dot{N}e}(\hat{a}_3)(\hat{s}_2)}, \alpha_{\dot{N}}(\hat{a}_3)(\hat{s}_2) \exp^{j\beta_{\dot{N}}(\hat{a}_3)(\hat{s}_2)} \rangle$

......................................................................................................................

......................................................................................................................

......................................................................................................................

$\hat{\omega}_{3q} = \langle \alpha_{\hat{P}}(\hat{a}_3)(\hat{s}_q) \exp^{j\beta_{\hat{P}}(\hat{a}_3)(\hat{s}_q)}, \alpha_{\dot{N}e}(\hat{a}_3)(\hat{s}_q) \exp^{j\beta_{\dot{N}e}(\hat{a}_3)(\hat{s}_q)}, \alpha_{\dot{N}}(\hat{a}_3)(\hat{s}_q) \exp^{j\beta_{\dot{N}}(\hat{a}_3)(\hat{s}_q)} \rangle$

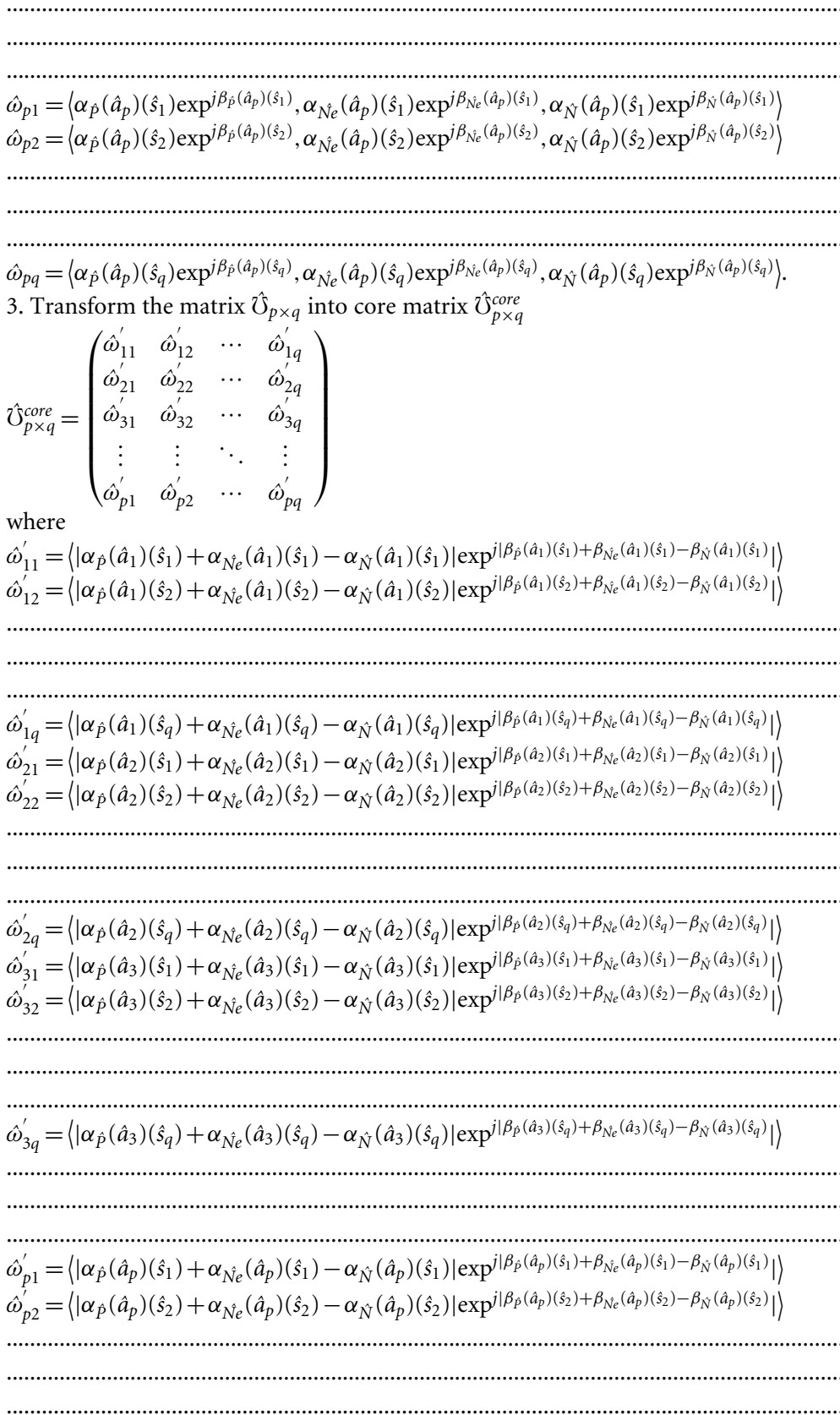

$$\hat{\omega}_{p1} = \left\langle \alpha_{\acute{P}}(\hat{a}_p)(\hat{s}_1)\exp^{j\beta_{\acute{P}}(\hat{a}_p)(\hat{s}_1)}, \alpha_{\grave{N}e}(\hat{a}_p)(\hat{s}_1)\exp^{j\beta_{\grave{N}e}(\hat{a}_p)(\hat{s}_1)}, \alpha_{\acute{N}}(\hat{a}_p)(\hat{s}_1)\exp^{j\beta_{\acute{N}}(\hat{a}_p)(\hat{s}_1)} \right\rangle$$

$$\hat{\omega}_{p2} = \left\langle \alpha_{\acute{P}}(\hat{a}_p)(\hat{s}_2)\exp^{j\beta_{\acute{P}}(\hat{a}_p)(\hat{s}_2)}, \alpha_{\grave{N}e}(\hat{a}_p)(\hat{s}_2)\exp^{j\beta_{\grave{N}e}(\hat{a}_p)(\hat{s}_2)}, \alpha_{\acute{N}}(\hat{a}_p)(\hat{s}_2)\exp^{j\beta_{\acute{N}}(\hat{a}_p)(\hat{s}_2)} \right\rangle$$

$$\hat{\omega}_{pq} = \left\langle \alpha_{\acute{P}}(\hat{a}_p)(\hat{s}_q)\exp^{j\beta_{\acute{P}}(\hat{a}_p)(\hat{s}_q)}, \alpha_{\grave{N}e}(\hat{a}_p)(\hat{s}_q)\exp^{j\beta_{\grave{N}e}(\hat{a}_p)(\hat{s}_q)}, \alpha_{\acute{N}}(\hat{a}_p)(\hat{s}_q)\exp^{j\beta_{\acute{N}}(\hat{a}_p)(\hat{s}_q)} \right\rangle.$$

3. Transform the matrix $\hat{\mho}_{p \times q}$ into core matrix $\hat{\mho}_{p \times q}^{core}$

$$\hat{\mho}_{p \times q}^{core} = \begin{pmatrix} \hat{\omega}'_{11} & \hat{\omega}'_{12} & \cdots & \hat{\omega}'_{1q} \\ \hat{\omega}'_{21} & \hat{\omega}'_{22} & \cdots & \hat{\omega}'_{2q} \\ \hat{\omega}'_{31} & \hat{\omega}'_{32} & \cdots & \hat{\omega}'_{3q} \\ \vdots & \vdots & \ddots & \vdots \\ \hat{\omega}'_{p1} & \hat{\omega}'_{p2} & \cdots & \hat{\omega}'_{pq} \end{pmatrix}$$

where

$$\hat{\omega}'_{11} = \left\langle |\alpha_{\acute{P}}(\hat{a}_1)(\hat{s}_1) + \alpha_{\grave{N}e}(\hat{a}_1)(\hat{s}_1) - \alpha_{\acute{N}}(\hat{a}_1)(\hat{s}_1)|\exp^{j|\beta_{\acute{P}}(\hat{a}_1)(\hat{s}_1) + \beta_{\grave{N}e}(\hat{a}_1)(\hat{s}_1) - \beta_{\acute{N}}(\hat{a}_1)(\hat{s}_1)|} \right\rangle$$

$$\hat{\omega}'_{12} = \left\langle |\alpha_{\acute{P}}(\hat{a}_1)(\hat{s}_2) + \alpha_{\grave{N}e}(\hat{a}_1)(\hat{s}_2) - \alpha_{\acute{N}}(\hat{a}_1)(\hat{s}_2)|\exp^{j|\beta_{\acute{P}}(\hat{a}_1)(\hat{s}_2) + \beta_{\grave{N}e}(\hat{a}_1)(\hat{s}_2) - \beta_{\acute{N}}(\hat{a}_1)(\hat{s}_2)|} \right\rangle$$

$$\hat{\omega}'_{1q} = \left\langle |\alpha_{\acute{P}}(\hat{a}_1)(\hat{s}_q) + \alpha_{\grave{N}e}(\hat{a}_1)(\hat{s}_q) - \alpha_{\acute{N}}(\hat{a}_1)(\hat{s}_q)|\exp^{j|\beta_{\acute{P}}(\hat{a}_1)(\hat{s}_q) + \beta_{\grave{N}e}(\hat{a}_1)(\hat{s}_q) - \beta_{\acute{N}}(\hat{a}_1)(\hat{s}_q)|} \right\rangle$$

$$\hat{\omega}'_{21} = \left\langle |\alpha_{\acute{P}}(\hat{a}_2)(\hat{s}_1) + \alpha_{\grave{N}e}(\hat{a}_2)(\hat{s}_1) - \alpha_{\acute{N}}(\hat{a}_2)(\hat{s}_1)|\exp^{j|\beta_{\acute{P}}(\hat{a}_2)(\hat{s}_1) + \beta_{\grave{N}e}(\hat{a}_2)(\hat{s}_1) - \beta_{\acute{N}}(\hat{a}_2)(\hat{s}_1)|} \right\rangle$$

$$\hat{\omega}'_{22} = \left\langle |\alpha_{\acute{P}}(\hat{a}_2)(\hat{s}_2) + \alpha_{\grave{N}e}(\hat{a}_2)(\hat{s}_2) - \alpha_{\acute{N}}(\hat{a}_2)(\hat{s}_2)|\exp^{j|\beta_{\acute{P}}(\hat{a}_2)(\hat{s}_2) + \beta_{\grave{N}e}(\hat{a}_2)(\hat{s}_2) - \beta_{\acute{N}}(\hat{a}_2)(\hat{s}_2)|} \right\rangle$$

$$\hat{\omega}'_{2q} = \left\langle |\alpha_{\acute{P}}(\hat{a}_2)(\hat{s}_q) + \alpha_{\grave{N}e}(\hat{a}_2)(\hat{s}_q) - \alpha_{\acute{N}}(\hat{a}_2)(\hat{s}_q)|\exp^{j|\beta_{\acute{P}}(\hat{a}_2)(\hat{s}_q) + \beta_{\grave{N}e}(\hat{a}_2)(\hat{s}_q) - \beta_{\acute{N}}(\hat{a}_2)(\hat{s}_q)|} \right\rangle$$

$$\hat{\omega}'_{31} = \left\langle |\alpha_{\acute{P}}(\hat{a}_3)(\hat{s}_1) + \alpha_{\grave{N}e}(\hat{a}_3)(\hat{s}_1) - \alpha_{\acute{N}}(\hat{a}_3)(\hat{s}_1)|\exp^{j|\beta_{\acute{P}}(\hat{a}_3)(\hat{s}_1) + \beta_{\grave{N}e}(\hat{a}_3)(\hat{s}_1) - \beta_{\acute{N}}(\hat{a}_3)(\hat{s}_1)|} \right\rangle$$

$$\hat{\omega}'_{32} = \left\langle |\alpha_{\acute{P}}(\hat{a}_3)(\hat{s}_2) + \alpha_{\grave{N}e}(\hat{a}_3)(\hat{s}_2) - \alpha_{\acute{N}}(\hat{a}_3)(\hat{s}_2)|\exp^{j|\beta_{\acute{P}}(\hat{a}_3)(\hat{s}_2) + \beta_{\grave{N}e}(\hat{a}_3)(\hat{s}_2) - \beta_{\acute{N}}(\hat{a}_3)(\hat{s}_2)|} \right\rangle$$

$$\hat{\omega}'_{3q} = \left\langle |\alpha_{\acute{P}}(\hat{a}_3)(\hat{s}_q) + \alpha_{\grave{N}e}(\hat{a}_3)(\hat{s}_q) - \alpha_{\acute{N}}(\hat{a}_3)(\hat{s}_q)|\exp^{j|\beta_{\acute{P}}(\hat{a}_3)(\hat{s}_q) + \beta_{\grave{N}e}(\hat{a}_3)(\hat{s}_q) - \beta_{\acute{N}}(\hat{a}_3)(\hat{s}_q)|} \right\rangle$$

$$\hat{\omega}'_{p1} = \left\langle |\alpha_{\acute{P}}(\hat{a}_p)(\hat{s}_1) + \alpha_{\grave{N}e}(\hat{a}_p)(\hat{s}_1) - \alpha_{\acute{N}}(\hat{a}_p)(\hat{s}_1)|\exp^{j|\beta_{\acute{P}}(\hat{a}_p)(\hat{s}_1) + \beta_{\grave{N}e}(\hat{a}_p)(\hat{s}_1) - \beta_{\acute{N}}(\hat{a}_p)(\hat{s}_1)|} \right\rangle$$

$$\hat{\omega}'_{p2} = \left\langle |\alpha_{\acute{P}}(\hat{a}_p)(\hat{s}_2) + \alpha_{\grave{N}e}(\hat{a}_p)(\hat{s}_2) - \alpha_{\acute{N}}(\hat{a}_p)(\hat{s}_2)|\exp^{j|\beta_{\acute{P}}(\hat{a}_p)(\hat{s}_2) + \beta_{\grave{N}e}(\hat{a}_p)(\hat{s}_2) - \beta_{\acute{N}}(\hat{a}_p)(\hat{s}_2)|} \right\rangle$$

$$\acute{\omega}'_{pq} = \left\langle |\alpha_{\acute{P}}(\hat{a}_p)(\hat{s}_q) + \alpha_{\grave{N}e}(\hat{a}_p)(\hat{s}_q) - \alpha_{\grave{N}}(\hat{a}_p)(\hat{s}_q)| \exp^{j|\beta_{\acute{P}}(\hat{a}_p)(\hat{s}_q) + \beta_{\grave{N}e}(\hat{a}_p)(\hat{s}_q) - \beta_{\grave{N}}(\hat{a}_p)(\hat{s}_q)|} \right\rangle.$$

4. Split the core matrix $\hat{\mho}^{core}_{p\times q}$ into core matrix for amplitude terms $\hat{\mho}^{core}_{amp}$ and core matrix for phase terms $\hat{\mho}^{core}_{pha}$ as given below

$$\hat{\mho}^{core}_{amp} = \begin{pmatrix} \hat{\omega}''_{11} & \hat{\omega}''_{12} & \cdots & \hat{\omega}''_{1q} \\ \hat{\omega}''_{21} & \hat{\omega}''_{22} & \cdots & \hat{\omega}''_{2q} \\ \hat{\omega}''_{31} & \hat{\omega}''_{32} & \cdots & \hat{\omega}''_{3q} \\ \vdots & \vdots & \ddots & \vdots \\ \hat{\omega}''_{p1} & \hat{\omega}''_{p2} & \cdots & \hat{\omega}''_{pq} \end{pmatrix}_{p\times q}$$

where $\hat{\omega}''_{lm} = \alpha(\hat{a}_l)(\hat{s}_m), l \in \{1,2,\ldots,p\} \& m \in \{1,2,\ldots,q\}$ and

$$\hat{\mho}^{core}_{pha} = \begin{pmatrix} \hat{\omega}'''_{11} & \hat{\omega}'''_{12} & \cdots & \hat{\omega}'''_{1q} \\ \hat{\omega}'''_{21} & \hat{\omega}'''_{22} & \cdots & \hat{\omega}'''_{2q} \\ \hat{\omega}'''_{31} & \hat{\omega}'''_{32} & \cdots & \hat{\omega}'''_{3q} \\ \vdots & \vdots & \ddots & \vdots \\ \hat{\omega}'''_{p1} & \hat{\omega}'''_{p2} & \cdots & \hat{\omega}'''_{pq} \end{pmatrix}_{p\times q}$$

where $\hat{\omega}'''_{lm} = \beta(\hat{a}_l)(\hat{s}_m), l \in \{1,2,\ldots,p\} \& m \in \{1,2,\ldots,q\}$.

5. Compute maximum decision values $\mathbb{D}^{max}_{amp}(\hat{s}_m)$, minimum decision values $\mathbb{D}^{min}_{amp}(\hat{s}_m)$ and score values $\mathfrak{S}_{amp}(\hat{s}_m)$ for each alternative $\hat{s}_m, m \in \{1,2,\ldots,q\}$ from matrix $\hat{\mho}^{core}_{amp}$. Similarly compute the same values $\mathbb{D}^{max}_{pha}(\hat{s}_m), \mathbb{D}^{min}_{pha}(\hat{s}_m)$ and $\mathfrak{S}_{pha}(\hat{s}_m)$ from matrix $\hat{\mho}^{core}_{pha}$ by using the following formulae:

$$\mathbb{D}^{max}_{amp}(\hat{s}_m) = \sum_{l=1}^{p} (1 - \alpha(\hat{a}_l)(\hat{s}_m))^2. \tag{1}$$

$$\mathbb{D}^{min}_{amp}(\hat{s}_m) = \sum_{l=1}^{p} (\alpha(\hat{a}_l)(\hat{s}_m))^2. \tag{2}$$

$$\mathfrak{S}_{amp}(\hat{s}_m) = \frac{\mathbb{D}^{max}_{amp}(\hat{s}_m) + \mathbb{D}^{min}_{amp}(\hat{s}_m)}{m}. \tag{3}$$

$$\mathbb{D}^{max}_{pha}(\hat{s}_m) = \sum_{l=1}^{p} (1 - \beta(\hat{a}_l)(\hat{s}_m))^2. \tag{4}$$

$$\mathbb{D}^{min}_{pha}(\hat{s}_m) = \sum_{l=1}^{p} (\beta(\hat{a}_l)(\hat{s}_m))^2. \tag{5}$$

$$\mathfrak{S}_{pha}(\hat{s}_m) = \frac{\mathbb{D}^{max}_{pha}(\hat{s}_m) + \mathbb{D}^{min}_{pha}(\hat{s}_m)}{m}. \tag{6}$$

(to understand the motivation behind this method, let $\rho$ be the Euclidean metric on $R^l$, $0 = (0, \dots, 0)^T \in R^l$, $1 = (1, \dots, 1)^T \in R^l$, and $\theta_j = \left(\theta_{1,\hat{u}_j}, \theta_{2,\hat{u}_j}, \dots, \theta_{l,\hat{u}_j}\right)^T \in R^l$. Thus $\mathfrak{S}(\hat{u}_j) = \left[\rho(\theta_j, 1)\right]^2 + \left[\rho(\theta_j, 0)\right]^2 (j = 1, 2, \dots, k)$.

6. Compute mean $\mathfrak{S}(\hat{s}_m)$ by using the following formula:

$$\mathfrak{S}(\hat{s}_m) = \frac{\mathbb{S}_{amp}(\hat{s}_m) + \mathbb{S}_{pha}(\hat{s}_m)}{2}. \tag{7}$$

7. Select the alternative with maximum $\mathfrak{S}(\hat{s}_m)$ as optimal recommendation.

The step-wise graphical exploration of Algorithm 1 is presented in Fig. 1.

## Implementation of proposed algorithm in real life scenario

This section is aimed to explain the proposed algorithm (*i.e.*, Algorithm 1) with the help of real life MADM-based scenario which is discussed in the following example.

**Example 4.1** Consider a real estate company "BUILDCO" (a hypothetical name) wants to start a new construction project. The managing director (MD) of the company is very much concerned about the substandard construction-related materials available in the market. Therefore the MD has decided to hire the services of a supplier in this regards. Two internal domestic committees $\mathfrak{A} = \{\hat{X}_1, \hat{X}_2\}$ and $\mathfrak{B} = \{\hat{X}_3, \hat{X}_4\}$ are constituted which have expert officers of the company. The committee $\mathfrak{A}$ is meant for initial screening of the suppliers and the committee $\mathfrak{B}$ is meant for further evaluation and final ranking of the suppliers. Three suppliers are short-listed by the committee $\mathfrak{A}$ which form the initial space of alternatives $\hat{\mathbb{S}} = \{\hat{s}_1, \hat{s}_2, \hat{s}_3\}$. With mutual understanding, the members of committee $\mathfrak{B}$ are agreed on attributes $\hat{a}_1 =$ quality cum reliability, $\hat{a}_2 =$ affordable cost and $\hat{a}_3 =$ service cum processing. The experts of committee $\mathfrak{B}$ provide their opinions for each alternative in set $\hat{\mathbb{S}}$ in terms of cpFSNs by considering the attributes. The complete evaluation of suppliers is accomplished by following the steps of the proposed Algorithm 1.

(1,2) A cpFSS $\hat{\Pi}$ is constructed which is presented in matrix notation $\hat{\mho}_{3 \times 3}$ as provided below

$$\hat{\mho}_{3\times3} = \begin{pmatrix} \hat{\omega}_{11} & \hat{\omega}_{12} & \hat{\omega}_{13} \\ \hat{\omega}_{21} & \hat{\omega}_{22} & \hat{\omega}_{23} \\ \hat{\omega}_{31} & \hat{\omega}_{32} & \hat{\omega}_{33} \end{pmatrix}$$

where

$$\hat{\omega}_{11} = \left\langle 0.08\, exp^{j2\pi(0.2)}, 0.4\, exp^{j2\pi(0.25)}, 0.03\, exp^{j2\pi(0.12)}\right\rangle,$$
$$\hat{\omega}_{12} = \left\langle 0.07\, exp^{j2\pi(0.21)}, 0.5\, exp^{j2\pi(0.4)}, 0.14\, exp^{j2\pi(0.12)}\right\rangle,$$
$$\hat{\omega}_{13} = \left\langle 0.06\, exp^{j2\pi(0.21)}, 0.6\, exp^{j2\pi(0.35)}, 0.01\, exp^{j2\pi(0.2)}\right\rangle,$$
$$\hat{\omega}_{21} = \left\langle 0.16\, exp^{j2\pi(0.31)}, 0.6\, exp^{j2\pi(0.45)}, 0.06\, exp^{j2\pi(0.21)}\right\rangle,$$
$$\hat{\omega}_{22} = \left\langle 0.09\, exp^{j2\pi(0.21)}, 0.06\, exp^{j2\pi(0.13)}, 0.11\, exp^{j2\pi(0.32)}\right\rangle,$$
$$\hat{\omega}_{23} = \left\langle 0.03\, exp^{j2\pi(0.11)}, 0.51\, exp^{j2\pi(0.5)}, 0.08\, exp^{j2\pi(0.21)}\right\rangle,$$
$$\hat{\omega}_{31} = \left\langle 0.07\, exp^{j2\pi(0.31)}, 0.42\, exp^{j2\pi(0.43)}, 0.05\, exp^{j2\pi(0.12)}\right\rangle,$$
$$\hat{\omega}_{32} = \left\langle 0.26\, exp^{j2\pi(0.36)}, 0.5\, exp^{j2\pi(0.53)}, 0.04\, exp^{j2\pi(0.12)}\right\rangle,$$
$$\hat{\omega}_{33} = \left\langle 0.08\, exp^{j2\pi(0.41)}, 0.34\, exp^{j2\pi(0.25)}, 0.03\, exp^{j2\pi(0.3)}\right\rangle.$$

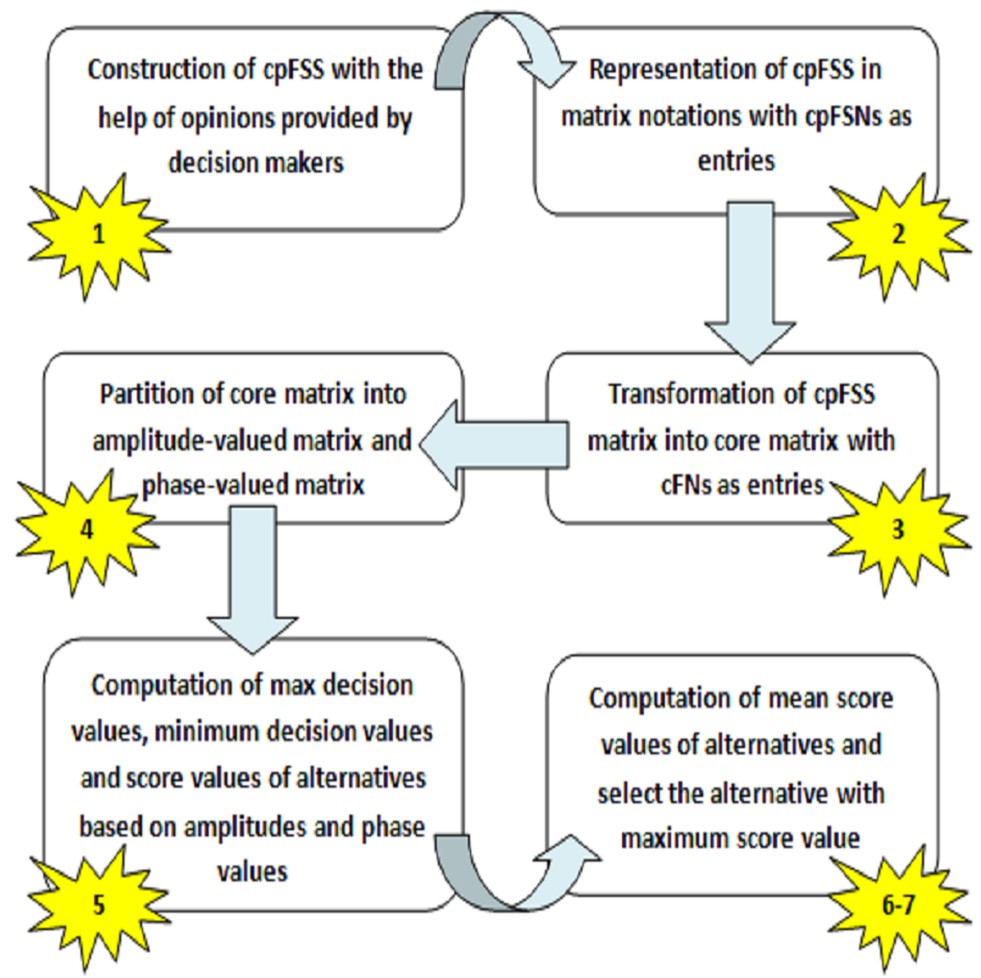

**Figure 1** Step-wise graphical representation of proposed algorithm.

(3) The matrix $\hat{U}_{3\times3}$ is transformed into core matrix $\hat{U}^{core}_{3\times3}$

$$\hat{U}^{core}_{3\times3} = \begin{pmatrix} \langle 0.35\,exp^{j2\pi(0.33)} \rangle & \langle 0.43,\,exp^{j2\pi(0.59)} \rangle & \langle 0.65,\,exp^{j2\pi(0.36)} \rangle \\ \langle 0.7,\,exp^{j2\pi(0.55)} \rangle & \langle 0.04,\,exp^{j2\pi(0.02)} \rangle & \langle 0.46,\,exp^{j2\pi(0.36)} \rangle \\ \langle 0.43,\,exp^{j2\pi(0.62)} \rangle & \langle 0.72,\,exp^{j2\pi(0.77)} \rangle & \langle 0.39,\,exp^{j2\pi(0.36)} \rangle \end{pmatrix}.$$

(4) The core matrix $\hat{U}^{core}_{p\times q}$ is partitioned into core matrix for amplitude terms $\hat{U}^{core}_{amp}$ and core matrix for phase terms $\hat{U}^{core}_{pha}$ as given below

$$\hat{U}^{core}_{amp} = \begin{pmatrix} 0.35 & 0.43 & 0.65 \\ 0.43 & 0.72 & 0.39 \end{pmatrix}_{3\times3},$$

and

$$\hat{U}^{core}_{pha} = \begin{pmatrix} 0.33 & 0.59 & 0.36 \\ 0.55 & 0.02 & 0.36 \\ 0.62 & 0.77 & 0.36 \end{pmatrix}_{3\times3}.$$

(5) Maximum decision values $\mathbb{D}^{max}_{amp}(\hat{s}_m)$, minimum decision values $\mathbb{D}^{min}_{amp}(\hat{s}_m)$ and score values $\mathbb{S}_{amp}(\hat{s}_m)$ for each alternative $\hat{s}_m, m \in \{1,2,3\}$ are computed

from matrix $\hat{\mho}_{amp}^{core}$. Similarly the same values $\mathbb{D}_{pha}^{max}(\hat{s}_m)$, $\mathbb{D}_{pha}^{min}(\hat{s}_m)$ and $\mathfrak{S}_{pha}(\hat{s}_m)$ are computed from matrix $\hat{\mho}_{pha}^{core}$ by using Eqs. (1)-(6) and are provided as

| $\hat{\mho}_{amp}^{core}$ | $\hat{s}_1$ | $\hat{s}_2$ | $\hat{s}_3$ |
|---|---|---|---|
| $\mathbb{D}_{amp}^{max}(\hat{s}_m)$ | 0.8699 | 1.3032 | 0.7754 |
| $\mathbb{D}_{amp}^{min}(\hat{s}_m)$ | 0.3393 | 0.5492 | 0.7038 |
| $\mathfrak{S}_{amp}(\hat{s}_m)$ | 0.4031 | 0.6175 | 0.4931 |

and

| $\hat{\mho}_{pha}^{core}$ | $\hat{s}_1$ | $\hat{s}_2$ | $\hat{s}_3$ |
|---|---|---|---|
| $\mathbb{D}_{pha}^{max}(\hat{s}_m)$ | 1.0266 | 1.5725 | 0.6069 |
| $\mathbb{D}_{pha}^{min}(\hat{s}_m)$ | 0.5866 | 0.4325 | 1.1069 |
| $\mathfrak{S}_{pha}(\hat{s}_m)$ | 0.5377 | 0.6683 | 0.5713 |

(6) The mean score $\mathfrak{S}(\hat{s}_m)$ is computed by using Eq. (7) and is given as

| | $\hat{s}_1$ | $\hat{s}_2$ | $\hat{s}_3$ |
|---|---|---|---|
| $\mathfrak{S}(\hat{s}_m)$ | 0.4704 | 0.6429 | 0.5322 |

(7) Since $\hat{s}_2$ has achieved the maximum score 0.6429 therefore $\hat{s}_2$ is recommended for the project. Hence the ranking of alternatives is $\hat{s}_2 > \hat{s}_3 > \hat{s}_1$.

## Sensitivity analysis-based discussion

Now sensitivity analysis of score values for suppliers are carried out in this part of the article to assess the stability of obtained results. For this purpose, we have employed different statistical tools like Pythagorean means and measures of dispersion to compute the approximate score value for the ranking of suppliers. Thus we discuss the following cases:

1. Geometric Mean: If we apply the idea of geometric mean for the computation of mean score, we have the equation

$$\mathfrak{S}(\hat{s}_m) = \sqrt{\mathbb{S}_{amp}(\hat{s}_m) \times \mathbb{S}_{pha}(\hat{s}_m)}. \qquad (8)$$

and by this equation, we get the following results:

| | $\hat{s}_1$ | $\hat{s}_2$ | $\hat{s}_3$ |
|---|---|---|---|
| $\mathfrak{S}(\hat{s}_m)$ | 0.4656 | 0.6424 | 0.5308 |

2.  Harmonic Mean: If we apply the idea of harmonic mean for the computation of mean score, we have the equation

$$\mathbb{S}(\hat{s}_m) = \frac{2}{\frac{1}{\mathbb{S}_{amp}(\hat{s}_m)} + \frac{1}{\mathbb{S}_{pha}(\hat{s}_m)}}. \tag{9}$$

and by this equation, we get the following results:

|  | $\hat{s}_1$ | $\hat{s}_2$ | $\hat{s}_3$ |
|---|---|---|---|
| $\mathbb{S}(\hat{s}_m)$ | 0.4608 | 0.6419 | 0.5293 |

3.  Variance: If we use the idea of variance, we the following results:

|  | $\hat{s}_1$ | $\hat{s}_2$ | $\hat{s}_3$ |
|---|---|---|---|
| $\mathbb{S}(\hat{s}_m)$ | 0.0045 | 0.0006 | 0.0015 |

4.  Standard Deviation: If we use the idea of standard deviation, we the following results:

|  | $\hat{s}_1$ | $\hat{s}_2$ | $\hat{s}_3$ |
|---|---|---|---|
| $\mathbb{S}(\hat{s}_m)$ | 0.0673 | 0.0254 | 0.0391 |

The readers will be able to determine how the score values affect the ranking or evaluation process of the suppliers by using Pythagorean means and measures of dispersion. The computed results and thus ranking of suppliers are presented in Table 1. From the Table 1 and Fig. 2, it can easily be observed that the ranking of $\hat{s}_3$ is consistent whereas the ranking of $\hat{s}_1$ and $\hat{s}_2$ are analogous to each other. In other words, $\hat{s}_2$ is more preferable than $\hat{s}_1$.

## Comparison based discussion

As discussed earlier, several researchers (*Erceg & Mularifović, 2019*; *Tamošaitiene et al., 2017*; *Chen & Su, 2022*; *Xiao, Chen & Li, 2012*; *Zulqarnain et al., 2021a*; *Zulqarnain et al., 2021b*; *Chang, 2019*; *Chatterjee, Mukherjee & Kar, 2018*; *Wang, Cheng & Huang, 2009*; *Junior, Osiro & Carpinetti, 2014*; *Büyüközkan & Çifçi, 2011*; *Büyüközkan & Göçer, 2017*; *Patra & Mondal, 2016*; *Agarwal, Biswas & Hanmandlu, 2013*; *García et al., 2013*; *Thao, 2021*; *Zhao et al., 2017*; *Khaleie, Fasanghari & Tavassoli, 2012*; *Liu & Wang, 2022*) have made rich contributions in SCMG and SSP with uncertainties; however, the contributions of scholars (*Zulqarnain et al., 2021b*; *Chang, 2019*; *Chatterjee, Mukherjee & Kar, 2018*; *Patra & Mondal, 2016*; *Agarwal, Biswas & Hanmandlu, 2013*) are observed to be the most relevant

**Table 1** Sensitivity analysis-based comparison.

| Techniques | $\mathfrak{S}(\acute{s}_1)$ | $\mathfrak{S}(\acute{s}_2)$ | $\mathfrak{S}(\acute{s}_3)$ | Ranking |
|---|---|---|---|---|
| Arithmetic Mean | 0.4704 | 0.6429 | 0.5322 | $\acute{s}_2 > \acute{s}_3 > \acute{s}_1$ |
| Geometric Mean | 0.4656 | 0.6424 | 0.5308 | $\acute{s}_2 > \acute{s}_3 > \acute{s}_1$ |
| Harmonic Mean | 0.4608 | 0.6419 | 0.5293 | $\acute{s}_2 > \acute{s}_3 > \acute{s}_1$ |
| Variance | 0.0045 | 0.0006 | 0.0015 | $\acute{s}_1 > \acute{s}_3 > \acute{s}_2$ |
| Standard Deviation | 0.0673 | 0.0254 | 0.0391 | $\acute{s}_1 > \acute{s}_3 > \acute{s}_2$ |

to the proposed study. Therefore, their computational findings are analyzed for comparison with the computed results of the proposed approaches. This kind of computation-based comparison is presented in Table 2. As in the proposed study, only three suppliers are evaluated and ranked; therefore, for the sake of comparison, the computation-based results of the first three suppliers are considered from the above-mentioned references, and the results of the remaining suppliers are omitted due to their irrelevancy to this study. From Table 2, it is clear that the proposed approach is more reliable and trustworthy due to consistent results, *i.e.*, very small values that usually lead to converging solutions with fewer error expectations. It is also pertinent to note that the ranking of suppliers is still consistent with the existing studies, even after considering complex settings.

Moreover, the above-mentioned references are inadequate with the following issues:

1. The periodicity of information is frequently faced while dealing with huge amounts of data, which may have a significant impact on the decision to make. Complex settings that address periodicity with the use of amplitude and phase terms are used to manage this type of problem.

2. Sometimes, while approximating any alternative based on relevant characteristics, the decision-makers seek to stay neutral. The awarding of a suitable neutral grade typically resolves this problem, increasing the decision system's adaptability and dependability.

3. It is important to employ simple and straightforward calculations to appeal to and assist the diverse community because computational complexity reduces the readability and understanding of any topic.

The idea of cpFSS is characterised in the current work in order to address all the concerns mentioned above. The picture fuzzy option is intended to capture the second issue, while its complicated component (amplitude and phase terms) is intended to handle the first. The third problem is similarly handled by applying quick calculations based on matrix operations.

In Table 3, the abbreviations MrF, NMrF, NMrGd, AmpTrm, PhasTrm, ParaT, ×, and ✓ are meant for membership function, non membership function, neutral membership grade, amplitude term, phase term, parameterization tool, invalid, and valid respectively.

## CONCLUSION

This study is the blend of two dependent phases; in the first phase a novel fuzzy soft set-like structure, complex picture fuzzy soft set (cpFSS), is investigated and its set theoretic operations are characterized with the support of examples. The cpFSS is more reliable and

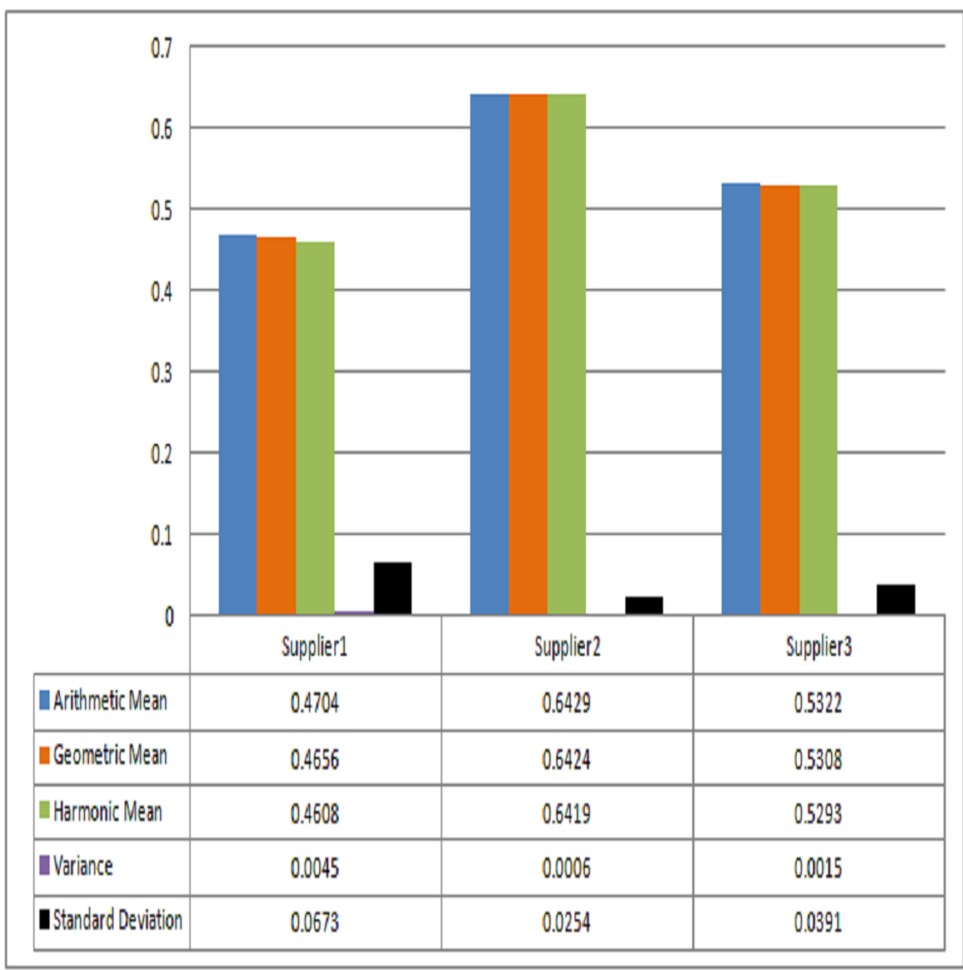

**Figure 2** Comparison of scores obtained by statistical tools.

**Table 2** Computations based comparison.

| References | $\mathfrak{S}(\acute{s}_1)$ | $\mathfrak{S}(\acute{s}_2)$ | $\mathfrak{S}(\acute{s}_3)$ | Ranking |
|---|---|---|---|---|
| *Zulqarnain et al. (2021b)* | 0.167 | 0.242 | 0.267 | $\acute{s}_3 > \acute{s}_2 > \acute{s}_1$ |
| *Chang (2019)* | 0.353 | 0.224 | 0.229 | $\acute{s}_1 > \acute{s}_3 > \acute{s}_2$ |
| *Chatterjee, Mukherjee & Kar (2018)* | 56.250 | 54.610 | 64.790 | $\acute{s}_3 > \acute{s}_1 > \acute{s}_2$ |
| *Patra & Mondal (2016)* | 0.943 | 0.918 | 0.895 | $\acute{s}_1 > \acute{s}_2 > \acute{s}_3$ |
| *Agarwal, Biswas & Hanmandlu (2013)* | 0.640 | 0.310 | 0.180 | $\acute{s}_1 > \acute{s}_2 > \acute{s}_3$ |
| Proposed 1 | 0.005 | 0.001 | 0.002 | $\acute{s}_1 > \acute{s}_3 > \acute{s}_2$ |
| Proposed 2 | 0.067 | 0.025 | 0.039 | $\acute{s}_1 > \acute{s}_3 > \acute{s}_2$ |

flexible as compared to cFS, cIFS, cpFS, cFSS and cIFSS as it has the capability to tackle the limitations of these models. In the second phase, a robust decision support system is constructed with the proposal of an MADM-based algorithm by using new matrix-based aggregations of cpFSS. Further, this algorithm is explained with the help of decision

**Table 3  Structural comparison.**

| References | MrF | NMrF | NMrGd | AmpTrm | PhasTrm | ParaT |
|---|---|---|---|---|---|---|
| *Zadeh (1965)* | ✓ | ✗ | ✗ | ✗ | ✗ | ✗ |
| *Atanassov (1986)* | ✓ | ✓ | ✗ | ✗ | ✗ | ✗ |
| *Cuong & Kreinovich (2013)* | ✓ | ✓ | ✓ | ✗ | ✗ | ✗ |
| *Ramot et al. (2002)* | ✓ | ✗ | ✗ | ✓ | ✓ | ✗ |
| *Molodtsov (1999)* | ✗ | ✗ | ✗ | ✗ | ✗ | ✓ |
| *Maji, Biswas & Roy (2001a)* | ✓ | ✗ | ✗ | ✗ | ✗ | ✓ |
| *Maji, Biswas & Roy (2001b)* | ✓ | ✓ | ✗ | ✗ | ✗ | ✓ |
| *Thirunavukarasu, Suresh & Ashokkumar (2017)* | ✓ | ✗ | ✗ | ✓ | ✓ | ✓ |
| *Kumar & Bajaj (2014)* | ✓ | ✓ | ✗ | ✓ | ✓ | ✓ |
| Proposed structure | ✓ | ✓ | ✓ | ✓ | ✓ | ✓ |

making scenario for the selection of suitable supplier to meet the material requirements of construction project. As far as the advantageous features of the proposed study are concerned, it is worth noting that the proposed study is more flexible and reliable as it has the capability to address most of the shortcomings encountered by existing FS-like structures. The major advantages of this study are outlined as:

1. It entitles amplitude and phase terms to cope with periodicity of information.
2. It entitles refusal grade to assist the experts for providing neutral opinions regarding the approximation of suppliers based on suitable attributes.
3. It provides an easy way of presenting mathematical computations in the form of matrix manipulations.
4. It generalizes most of the existing FS-like structures that proves its flexibility.

Some significant managerial implications regarding SSP with uncertain structure are risk management, contingency planning, technological integration, and continuous improvement and learning. By addressing these managerial implications and integrating with this study, companies can navigate the complexities associated with SSP under uncertainties, enhancing supply chain resilience, and maintaining a competitive advantage in the marketplace.

### Funding
The authors received no funding for this work.

### Competing Interests
The authors declare that there are no competing interests.

### Author Contributions
- Ali Asghar conceived and designed the experiments, performed the experiments, analyzed the data, prepared figures and/or tables, authored or reviewed drafts of the article, and approved the final draft.

- Khuram A. Khan conceived and designed the experiments, performed the experiments, analyzed the data, performed the computation work, authored or reviewed drafts of the article, and approved the final draft.
- Marwan A. Albahar performed the experiments, performed the computation work, prepared figures and/or tables, authored or reviewed drafts of the article, data curation, and approved the final draft.
- Abdullah Alammari conceived and designed the experiments, analyzed the data, authored or reviewed drafts of the article, and approved the final draft.

### Data Deposition

The raw measurements and codes are available in the Supplementary Files.

### Supplemental Information

Supplemental information for this article can be found online at http://dx.doi.org/10.7717/peerj-cs.1540#supplemental-information.

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
