# Peer review of "An optimized multi-attribute decision-making approach to construction supply chain management by using complex picture fuzzy soft set"

_PeerJ Computer Science, doi:10.7717/peerj-cs.1540_

## Round 0.1 · original submission · Major Revisions

The manuscript should be revised according to the suggestions of the reviewers.

Reviewer 1 ·

Basic reporting

The manuscript An optimized multi attribute decision making approach to construction supply chain management by using complex picture fuzzy soft set falls within the scope of the journal. The authors are familiar with the fields covered in the paper and have proposed strong methodology.
The paper has all elements to be considered in one prestigious journal.

Experimental design

Well performed.

Validity of the findings

Validity of the results through sensitivity analysis has been provided.

Additional comments

- Results should be included in the abstract
- A lot of studies in the introduction section have been only listed, but not elaborated properly. I propose that a separate section named Background or Literature review should be created. Some recently published articles should be added as: 1) Saha, A., Reddy, J., & Kumar, R. (2022). A fuzzy similarity based classification with Archimedean-Dombi aggregation operator. J. Intell Manag. Decis, 1(2), 118-127.
2) Chen, L. & Su, S. (2022). Optimization of the Trust Propagation on Supply Chain Network Based on Blockchain Plus, J. Intell. Manag. Decis., 1(1), 17-27. https://doi.org/10.56578/jimd010103
- Section Preliminaries should be extended with more explanations.
- More advantages of your approach should be elaborated.
- Should be created discussion section with shown clear contributions, limitations, and implications.
- The conclusion section must be extended.

Reviewer 2 ·

Basic reporting

The title of the paper is in correspondence with its content. In my opinion and experience with this topic authors need to add hyphens in the way - An optimized multi-attribute decision-making approach to construction supply chain management by using complex picture fuzzy soft set
The summary and conclusion correspond to the essence of the work.
The use of terminology is correct and it complies with applicable standards.

Experimental design

The paper presents a significant problem from the domain of logistics and highlights the difficulties that can affect the complexity of making a decision on the choice of supplier (line 40).
I suggest that the authors present in the introductory part, through a review of the literature, the existing ways of solving the supplier selection problem (SSP) applied so far (line 59) and highlight the most frequently used factors that are partially listed (line 55), and so far used.

Validity of the findings

In Conclusion, the authors said: The cpFSS is more reliable and flexible as compared to cFS, cIFS, 255 cpFS, cFSS, and cIFSS as it has the capability to tackle the limitations of these models.
I suggest that the authors briefly present the limitations of the mentioned existing models that the new model overcomes.

Additional comments

no comment

---

## Round 0.2 · accepted · Accept

The paper can be accepted.

Reviewer 1 ·

Basic reporting

...

Experimental design

Well performed.

Validity of the findings

Validity of the results through sensitivity analysis has been provided.

Additional comments

- Results should be included in the abstract
The authors have modified the abstract accordingly.
- A lot of studies in the introduction section have been only listed, but not elaborated properly. I propose that a separate section named Background or Literature review should be created. Some recently published articles should be added as: 1) Saha, A., Reddy, J., & Kumar, R. (2022). A fuzzy similarity based classification with Archimedean-Dombi aggregation operator. J. Intell Manag. Decis, 1(2), 118-127.
2) Chen, L. & Su, S. (2022). Optimization of the Trust Propagation on Supply Chain Network Based on Blockchain Plus, J. Intell. Manag. Decis., 1(1), 17-27. https://doi.org/10.56578/jimd010103
This task has been fulfilled.
- Section Preliminaries should be extended with more explanations.
Done.
- More advantages of your approach should be elaborated.
- Should be created discussion section with shown clear contributions, limitations, and implications.
Done
- The conclusion section must be extended.
Done

Reviewer 2 ·

Basic reporting

no comment

Experimental design

no comment

Validity of the findings

no comment

Additional comments

The authors adequately responded to all the suggestions and comments of the reviewers and improved their paper